# Twin pregnancy and perinatal outcomes: Data from 'Birth in Brazil Study'

**Ana Paula Esteves-Pereira**[1]*, **Antônio José Ledo Alves da Cunha**[2], **Marcos Nakamura-Pereira**[3], **Maria Elisabeth Moreira**[3], **Rosa Maria soares madeira Domingues**[4], **Elaine Fernandes Viellas**[1], **Maria do Carmo Leal**[1], **Silvana Granado nogueira da Gama**[1]

1 Department of Epidemiology and Quantitative Methods in Health, Sérgio Arouca National School of Public Health, Oswaldo Cruz Foundation, Rio de Janeiro, RJ, Brazil, 2 Medical School of Federal University of Rio de Janeiro, Rio de Janeiro, RJ, Brazil, 3 National Institute of Women, Children and Adolescents Health Fernandes Figueira, Oswaldo Cruz Foundation, Rio de Janeiro, RJ, Brazil, 4 National Institute of Infectious Diseases, Oswaldo Cruz Foundation, Rio de Janeiro, RJ, Brazil

* ana.pep@gmail.com

**Data Availability Statement:** All relevant data are within the manuscript and its Supporting Information files.

## Abstract

### Background

Twin pregnancies account for 0.5–2.0% of all gestations worldwide. They have a negative impact on perinatal health indicators, mainly owing to the increased risk for preterm birth. However, population-based data from low/middle income countries are limited. The current paper aims to understand the health risks of twins, compared to singletons, amongst late preterms and early terms.

### Methods

Data is from "Birth in Brazil", a national inquiry into childbirth care conducted in 2011/2012 in 266 maternity hospitals. We included women with a live birth or a stillborn, and excluded births of triplets or more, totalling 23,746 singletons and 554 twins. We used multiple logistic regressions and adjusted for potential confounders.

### Results

Twins accounted for 1.2% of gestations and 2.3% of newborns. They had higher prevalence of low birth weight and intrauterine growth restriction, when compared to singletons, in all gestational age groups, except in the very premature ones (<34 weeks). Amongst late preterm's, twins had higher odds of jaundice (OR 2.7, 95% CI 1.8–4.2) and antibiotic use (OR 1.8, 95% CI 1.1–3.2). Amongst early-terms, twins had higher odds of oxygen therapy (OR 2.7, 95% CI 1.3–5.9), admission to neonatal intensive care unit (OR 3.1, 95% CI 1.5–6.5), transient tachypnoea (OR 3.7, 95% CI 1.5–9.2), jaundice (OR 2.8, 95% CI 1.3–5.9) and antibiotic use (OR 2.2, 95% CI 1.14.9). In relation to birth order, the second-born infant had an elevated likelihood of jaundice, antibiotic use and oxygen therapy, than the first-born infant.

**Funding:** The Birth in Brazil Study was funded by the National Council for Scientific and Technological Development (CNPq); National School of Public Health, Oswaldo Cruz Foundation (INOVA Project); and Foundation for supporting Research in the State of Rio de Janeiro (FAPERJ). The funders had no role in study design, data collection and analysis, decision to publish, or preparation of the manuscript.

**Competing interests:** The authors have declared that no competing interests exist.

## Conclusion

Although strongly mediated by gestational age, an independent risk remains for twins for most neonatal morbidities, when compared to singletons. These disadvantages seem to be more prominent in early-term newborns than in the late preterm ones.

## Background

Twin births rates vary considerably around the world, according to country development, from less than 1% of all births in Asian South and Southeast [1] to more than 3% in the USA [2] and France [3]. Variations also exist according to race and ethnicity [1,2]. Multiple birth rates have been rising since the 1970s in developed countries [4,5]. Over three decades (1980–2009), the twin birth rate rose by 76% in USA, increasing more than 2% per year from 1980 through 2004 and reaching 3.3% of all births in 2009 [6]. The increasing maternal age after the emergence of new contraception and infertility treatments are the main causes of the increase in twin pregnancy rates [4,5,7–9].

In Brazil, according to the National Live Births Database (SINASC), there is also an increasing trend in twin birth rates, from 1.7% to 2.0% in the sixteen years since 2000 [10]. This trend was corroborated by a study in a Southern city of Brazil where multiple birth rates rose from 1.9% to 2.5% in the period 1994–2005 [11]. The prevalence of twin gestations is higher in the Brazilian regions with higher human development index (HDI). Moreover, mothers of twins are older and show higher levels of education [12].

Twin births have a negative impact on perinatal health indicators, since twins have a higher risk of perinatal mortality [12–14], especially due to higher preterm birth rates [5]. In low and middle income countries, early neonatal mortality is seven times higher among twins [14]. Even when we adjust these figures according to birth weight, the chance of early neonatal mortality in twins is three times higher than for single births. Several investigations have found that neonatal mortality and morbidity of the second twin are higher than the first [12,15,16]. Studies also point to the conflicting results on the safest mode of delivery for the second twin, particularly for non-cephalic second twins [16–18].

Moreover, maternal mortality and morbidity are higher in cases of twin births when compared with single pregnancies [13,19,20]. A large study conducted mainly in low- and middle-income countries has found that mothers of twins were three times more at risk of maternal near miss. The same group of women were considered four times more at risk of maternal mortality, particularly due to postpartum haemorrhage and hypertensive disorder [19].

In Brazil, as in several developing countries, surveys of twin pregnancies with country-specific data are limited. Most studies are hospital-based and consider a limited number of hospitals [21], or they are based on secondary data [11,12]. 'Birth in Brazil' was the first perinatal survey in Brazil to provide primary data comprising national and regional representative samples [22]. The study provides plentiful resource with which we can evaluate the magnitude of adverse outcomes in twin pregnancies in Brazil.

The objective of the current paper is to understand the health risks of giving birth to twins at different gestational ages. Subsequent analysis will explore the relationship between birth order and maternal and perinatal outcomes in cases of twin births.

## Methodology

### Database

The present manuscript is based on data from 'Birth in Brazil', a nationwide population-based survey on gestation and delivery, performed between February 2011 and October 2012. The

sample was selected in three stages. The first stage was to select hospitals with 500 or more births / year, stratified by the country's five macro-regions, location (capital or non-capital) and type of hospital (private, public and mixed). The second consisted of identifying the number of days needed to interview 90 puerperae (minimum of seven days in each hospital). The third consisted of selecting appropriate puerperae. 266 hospitals were sampled in total. Further details of sample design are available in Vasconcellos et al (2014) [23].

## Inclusion and exclusion criteria

The study included 24,035 women who were admitted to maternity wards at the time of delivery, along with their newborns of any gestational age and weight, and stillbirths with birth weight ≥ 500 g and / or gestational age ≥ 22 weeks' gestation. Of these, 23,746 had a single birth (23,746 newborns), 277 had twins (554 newborns) and 12 had triplets (36 newborns). For the present analysis, we have excluded deliveries of triplets. We thus sampled a total of 24,023 puerperae and 24,300 newborns.

## Predictor variables

The main predictor variable was twin pregnancy, compared to single pregnancies, stratified into four gestational age groups: <34 weeks (preterm), 34–36 weeks (late preterm), 37 and 38 weeks (early term) and ≥39 weeks (full term). As a secondary predictor variable, we analysed the birth order of twin infants (first-born or second-born).

## Outcomes

We analysed the following outcomes: low birthweight (<2500g); intrauterine growth restriction [IUGR] (below the tenth and third percentiles); resuscitation of the newborn in the delivery room (positive pressure ventilation, orotracheal intubation, cardiac massage or use of drugs); oxygen therapy (oxygen Hood, continuous positive airway pressure (CPAP) or mechanical ventilation); use of antibiotics at any time during hospitalisation; admission to the Neonatal Intensive Care Unit (NICU); transient tachypnoea of the newborn; hypoglycaemia in the first 48 hours of life; phototherapy in the first 72 hours of life (jaundice); severe neonatal outcomes; and severe maternal outcomes. The severe neonatal outcome included neonatal near misses, according to the WHO classification [24]. It also included early and late foetal and neonatal deaths. The severe maternal outcome included maternal near misses, according to the WHO classification [25], and maternal deaths occurring in the puerperal period. Appropriate birth weight for gestational age was assessed by Intergrowth-21st intrauterine growth curves, considering birth weight <10th percentile as small for gestational age (SGA) and birth weight <3th percentile as intrauterine growth restriction (IUGR) [26].

## Covariables

We identified the following variables as potential confounding factors: macroregion (North, Northeast, South, Southeast, Midwest), type of payment for childbirth (public or private); maternal age (<20, 20–34, ≥35); years of maternal schooling (≤7, 8–10, 11–14, ≥15); and gestational age upon delivery (in gestational weeks).

## Data collection

We collected data on the socioeconomic characteristics of the women (age, ethnicity, schooling, economic class, presence of a companion, and employment status) through face-to-face interviews with puerperal women. Data on obstetric history and maternal medical conditions

were collected from maternal hospital records and prenatal cards. We recorded all neonatal outcomes from hospital records of newborns.

We gathered information on gestational age upon delivery primarily from results of ultrasound examinations, performed between seven and thirteen gestational weeks. We collected this information from both the original ultrasound examination and prenatal cards, maternal hospital and newborn records. 74% of women had gestational age at birth classified by this method and, in the absence of ultrasonographic estimates, gestational age was based on the information reported by the woman in the interview (23%) and, finally, on the date of the last menstrual period (1%) and the 50% percentile of weight for gestational age at birth (2%) [27].

## Statistical analysis

Post-hoc calculations show that with a significance level of 5%, the twin sample (554 newborns) would have 80% power to detect an increased risk corresponding to an OR of $\geq 2$ for neonatal outcomes, with a prevalence of 5%. However, gestational age categories of 34–36 weeks (with 248 newborns) and 37–38 weeks (with 164 newborns) would have 80% power to detect an increased risk corresponding to an OR of $\geq 2.5$ and $\geq 3.0$, respectively, for neonatal outcomes with a prevalence of 5%.

We analysed the differences in the characteristics of postpartum women, as well as absolute differences in neonatal outcomes according to twin births, using the $\chi 2$ test. Using multiple non-conditional logistic regressions, stratified by the four gestational age categories ($<$34, 34–36, 37 and 38, $\geq$39), we analysed neonatal outcomes associated with twin births in comparison to single births. We analysed neonatal outcomes associated with birth order by means of multiple non-conditional logistic regression. In both analyses, we estimated the odds ratios (OR), adjusted odds ratios (adj. OR) and their respective 95% confidence intervals.

For all outcomes, macro-region, type of birth payment, maternal age, maternal schooling years and gestational age, we performed adjusted analysis. Gestational age at delivery was used as an adjustment variable in the model in complete gestational weeks, as there was a higher proportion of twins than single births at lower gestational ages, even within the pre-determined categories. We accounted for the complex sample design in all statistical analyses. We adopted a significance level of 5% for all analyses. For this research, we used the statistical programme SPSS V.20.0.

## Ethical considerations

This study was approved by the Research Ethics Committee of ENSP / FIOCRUZ under number 92/2010. Measures have been taken to ensure the privacy and confidentiality of data collected from participants. Informed consent was obtained prior to interviews with the use of an informed consent form.

## Results

Women giving birth to twins shared several defining characteristics, including higher usage of private healthcare services, older age ($\geq$ 35 years), and a greater prevalence of hypertensive disorders (chronic and gestational arterial hypertension, preeclampsia and HELLP syndrome) when compared to single birth mothers.

Among women giving birth to twins, the onset of labour was mostly provider-initiated, with almost no labour induction performed. Caesarean sections were performed in 84% of twin births and, amongst them, 61.5% were elective (antepartum) CS. Gestational age ranges in twin births also differed significantly from figures observed in single births (Table 1 and

**Table 1. Sociodemographic characteristics, medical conditions, mode of birth and gestational age among mothers of twins and singletons.**

| | Mothers of Twins (n = 277) | Mothers of Singletons (n = 23,746) | P-value* |
|---|---|---|---|
| **Total** | n (%) | n (%) | |
| **Region** | | | |
| North | 30 (10.8) | 2,291 (9.6) | 0.415 |
| Northeast | 92 (33.2) | 6,845 (28.8) | |
| Southeast | 96 (34.7) | 10,108 (42.6) | |
| South | 35 (12.6) | 2,959 (12.5) | |
| Midwest | 24 (8.7) | 1,543 (6.5) | |
| **Type of payment for bith** | | | |
| Public | 193 (69.7) | 19,074 (80.3) | 0.007 |
| Private | 84 (30.3) | 4,672 (19.7) | |
| **Age** | | | |
| 12 to 19 | 36 (13.0) | 4,543 (19.1) | <0.001 |
| 20 to 34 | 188 (67.9) | 16,709 (70.4) | |
| ≥ 35 | 53 (19.1) | 2,494 (10.5) | |
| **Ethnicity** | | | |
| White | 107 (38.6) | 8,002 (33.7) | 0.381 |
| Black | 19 (6.9) | 2,049 (8.6) | |
| Brown/ mixed | 151 (54.5) | 13,695 (57.7) | |
| **Years of schooling** | | | |
| ≤ 7 | 67 (24.2) | 6,349 (26.7) | 0.197 |
| 8 to 10 | 60 (21.7) | 6,125 (25.8) | |
| ≥ 11 | 150 (54.2) | 11,272 (47.5) | |
| **Economic class (n = 266 and 23,311)** | | | |
| D + E | 66 (24.8) | 5,521 (23.7) | 0.601 |
| C | 143 (53.8) | 12,123 (52.0) | |
| A + B | 57 (21.4) | 5,667 (24.3) | |
| **Marital status (n = 276 and 23,723)** | | | |
| Do not live with a companion | 42 (15.2) | 4,454 (18.8) | 0.208 |
| Live with a companion | 234 (84.8) | 19,269 (81.2) | |
| **Work (n = 273 and 23,646)** | | | |
| No | 161 (59) | 14,145 (59.8) | 0.823 |
| Yes | 112 (41) | 9,501 (40.2) | |
| **Previous births** | | | |
| Nuliparous | 123 (44.4) | 11,107 (46.8) | 0.310 |
| 1 to 2 | 115 (41.5) | 10,178 (42.7) | |
| ≥ 3 | 39 (14.2) | 2,460 (10.4) | |
| **Medical conditions (n = 276 and 23,611)** | | | |
| Chronic hypertension | 13 (4.7) | 617 (2.6) | 0.150 |
| Hypertensive disorders[a] | 51 (18.5) | 2,584 (10.9) | 0.003 |
| Eclampsia | 3 (1.1) | 135 (0.6) | 0.387 |
| Preexisting Diabetes | 6 (2.2) | 244 (1.0) | 0.196 |
| Gestational Diabetes | 27 (9.7) | 1,955 (8.2) | 0.471 |
| Other severe chronic diseases[b] | 5 (1.8) | 260 (1.1) | 0.256 |
| **Onset of labour (n = 276 and 23,611)** | | | |
| Spontaneous | 132 (47.8) | 13,514 (57.2) | <0.001 |
| Provider-initiated | 144 (52.2) | 10,097 (42.8) | |
| Elective (antepartum) CS | 142 (98.6) | 9,158 (90.7) | <0.001 |
| Successful labour induction | 2 (1.4) | 939 (9.3) | |

(Continued)

**Table 1.** (Continued)

|  | Mothers of Twins (n = 277) | Mothers of Singletons (n = 23,746) | P-value* |
|---|---|---|---|
| **Total** | **n (%)** | **n (%)** |  |
| **Mode of Birth** |  |  |  |
| Vaginal | 46 (16.3) | 11,545 (48.5) | <0.001 |
| Caesarean | 231 (83.7) | 12,201 (51.5) |  |
| Elective (antepartum) CS | 142 (61.5) | 9,158 (75.1) | 0.003 |
| Emergency (intrapartum) CS | 89 (38.5) | 3,043 (24.9) |  |
| **Gestational age in weeks** |  |  |  |
| < 34 | 50 (18.1) | 849 (3.6) | <0.001 |
| 34–36 | 124 (44.8) | 1,832 (7.7) |  |
| 37–38 | 82 (29.6) | 8,292 (34.9) |  |
| ≥ 39 | 21 (7.6) | 12,773 (53.8) |  |

* Chi square χ2 test.

a Gestational hypertension, pre-eclampsia and HELLP syndrome.

b Chronic cardiac diseases (other than HD), chronic renal diseases and auto-imune diseases.

S1 Fig). We detail the onset of labour and mode of birth in twin and singleton newborns, by gestational age groups, in S1 Table.

Newborn twins showed lower birthweights and a higher prevalence of restricted intrauterine growth, when compared to newborn single births. The results were independent of which percentile was considered (either tenth or third percentile) (Table 2). Of the 277 pairs of twins, 21% (59 pairs) were of the same sex and 20% (55 pairs) showed a difference in birthweights that exceeded 20% (data nor shown).

Newborn twins presented a larger risk of presenting neonatal outcomes, such as oxygen therapy, admission to neonatal intensive care unit, transient tachypnoea, jaundice and antibiotic use. Nevertheless, the absolute difference between twin and single newborns was more noticeable in late preterm infants (Table 2).

Among late preterm infants, for every one-hundred infants born, around 48% of newborn twins showed neonatal near miss/foetal mortality/neonatal mortality, compared to 28% of the same outcomes among single newborns. Equally, 30% of twins showing signs of jaundice were subjected to phototherapy, compared to 11% in single newborns; 29% of twins were admitted to an intensive care unit (ICU), compared to 18% in single newborns; and antibiotics were used to treat 20% of newborn twins, compared to 12% in single newborns (Table 2).

For early term infants, absolute differences between newborn twins and single-births were of a lesser magnitude, varying between 0.9% and 6.5%. Yet statistical significance was noted in a greater number of outcomes (Table 2).

In cases of extreme prematurity, the difference in the application of supplemental oxygen between twins and single newborns was more telling: 91% vs. 76%, respectively. Meanwhile, the difference in levels of admission to an ICU was comparable to figures observed in late preterm infants. Although, the overall proportion of extremely premature infants admitted to an ICU was far greater (Table 2).

In adjusted analysis, late preterm newborn twins showed a greater likelihood of being admitted to an ICU, receiving treatment with antibiotics, neonatal near miss, and undergoing phototherapy, with OR varying between 1.6 (CI 1.0–2.7) and 4.1 (CI 1.2–3.8). For early term infants, newborn twins were more likely to receive antibiotics, supplemental oxygen, phototherapy, to be admitted to an ICU, and to suffer from transient tachypnoea and

**Table 2. Prevalence of neonatal and maternal outcomes in twins and singletons by gestational age groups.**

| | Preterm (< 34 weeks) | | | | Late preterm (34–36 weeks) | | | | Early term (37–38 weeks) | | | | Full term (> = 39 weeks) | | | |
|---|---|---|---|---|---|---|---|---|---|---|---|---|---|---|---|---|
| | Twins (n = 100) n (%) | Singletons (n = 848) n (%) | Absolute diference (%) | P-value* | Twins (n = 248) n (%) | Singletons (n = 1,832) n (%) | Absolute diference (%) | P-value* | Twins (n = 164) n (%) | Singletons (n = 8,293) n (%) | Absolute diference (%) | P-value* | Twins (n = 42) n (%) | Singletons (n = 12,773) n (%) | Absolute diference (%) | P-value* |
| **Birth weight** | | | | | | | | | | | | | | | | |
| Mean (SD) | 1457g (559g) | 1515g (625g) | 58g | <0.001 | 2215g (350g) | 2665g (469g) | 450g | <0.001 | 2639g (341g) | 3126g (418g) | 487g | <0.001 | 2710g (317g) | 3354g (431g) | 644g | <0.001 |
| Low birth weight | 97 (96.7) | 785 (92.6) | 4.1 | 0.492 | 198 (82.0) | 624 (34.1) | 47.9 | <0.001 | 46 (29.1) | 457 (5.5) | 23.6 | <0.001 | 11 (26.3) | 250 (2.0) | 24.3 | <0.001 |
| SGA | 31 (31.7) | 206 (27.5) | 4.2 | 0.396 | 94 (39.0) | 165 (9.0) | 30.0 | <0.001 | 50 (31.9) | 434 (5.2) | 26.7 | <0.001 | 22 (52.4) | 1020 (8.0) | 44.4 | <0.001 |
| IUGR | 20 (19.9) | 138 (18.4) | 1.5 | 0.210 | 45 (18.8) | 93 (5.1) | 13.7 | 0.001 | 21 (13.6) | 158 (1.9) | 11.7 | <0.001 | 15 (35.2) | 457 (2.8) | 32.4 | <0.001 |
| **Neonatal and maternal outcomes** | | | | | | | | | | | | | | | | |
| Resuscitation | 46 (45.4) | 433 (51.1) | -5.7 | 0.531 | 24 (9.9) | 226 (12.3) | -2.4 | 0.327 | 9 (5.8) | 409 (4.9) | 0.9 | 0.700 | 4 (9.6) | 701 (5.5) | 4.1 | 0.402 |
| Oxygen therapy | 92 (91.4) | 642 (75.7) | 15.7 | 0.014 | 55 (22.8) | 344 (18.8) | 4.0 | 0.411 | 14 (8.9) | 261 (3.1) | 5.8 | 0.004 | 0 (0.0) | 291 (2.3) | -2.3 | 0.547 |
| Antibiotic use | 66 (65.8) | 531 (62.6) | 3.2 | 0.724 | 49 (20.3) | 230 (12.5) | 7.8 | 0.038 | 8 (4.8) | 173 (2.1) | 2.7 | 0.032 | 1 (2.2) | 250 (1.9) | 0.3 | 0.277 |
| Neonatal ICU admission | 87 (86.3) | 637 (75.1) | 11.2 | 0.060 | 70 (28.9) | 336 (18.3) | 10.6 | 0.009 | 15 (9.4) | 238 (2.9) | 6.5 | <0.001 | 0 (0.0) | 286 (2.2) | -2.2 | 0.556 |
| Transient tachypnea | 36 (35.9) | 221 (26.1) | 9.8 | 0.106 | 33 (13.6) | 199 (10.9) | 2.7 | 0.442 | 12 (7.4) | 158 (1.9) | 5.5 | 0.001 | 1 (2.2) | 195 (1.5) | 0.7 | 0.168 |
| Hypoglycemia | 15 (14.8) | 83 (9.8) | 5.0 | 0.247 | 11 (4.5) | 89 (4.9) | -0.4 | 0.709 | 7 (4.4) | 70 (0.8) | 3.6 | 0.001 | 0 (0.0) | 42 (0.3) | -0.3 | 0.806 |
| Phototherapy (jaundice) | 62 (61.2) | 371 (43.8) | 17.4 | 0.039 | 73 (30.2) | 205 (11.2) | 19.0 | <0.001 | 14 (8.8) | 236 (2.8) | 6.0 | <0.001 | 2 (4.5) | 268 (2.1) | 2.4 | 0.001 |
| Stillbirth | 5 (5.0) | 173 (20.4) | -15.4 | 0.001 | 0 (-) | 36 (1.9) | NA | NA | 0 (-) | 26 (0.3) | NA | NA | 1 (2.5) | 28 (0.2) | NA | NA |
| Neonatal death | 25 (25.0) | 158 (18.6) | 6.4 | 0.320 | 1 (0.4) | 35 (1.8) | NA | NA | 1 (0.6) | 21 (0.2) | NA | NA | 0 (-) | 27 (0.2) | NA | NA |
| Severe Neo. Out. Index [a] | 99 (97.7) | 798 (94.1) | 3.6 | 0.181 | 120 (48.6) | 517 (28.2) | 20.4 | 0.014 | 23 (14.0) | 678 (8.2) | 5.8 | 0.054 | 5 (12.7) | 859 (6.7) | 6.0 | 0.278 |
| Severe Mat. Out.Index [b] | 1 (1.0) | 59 (7.0) | -6.0 | <0.001 | 5 (2.0) | 50 (2.7) | -0.7 | 0.592 | 2 (1.2) | 83 (1.0) | 0.2 | 0.822 | 0 (0.0) | 72 (0.6) | -0.6 | 0.773 |

* T-test for birthweight mean and Chi square χ2 test for the categorical variables.

a Neonatal nearmiss, stillbirth or neonatal death.

b Maternal nearmiss or maternal death.

NA: Not applicable due to very few subjects with the outcome.

hypoglycaemia, with OR varying between 2.5 (CI 1.1–5.6) and 6.2 (CI 1.9–20.0). Twin births were neither associated with maternal near miss nor maternal mortality (Table 3).

When analysing birth order in twins, we found that the second-born twin showed a greater prevalence of the outcomes studied in this research, compared to the first-born twin. After adjusting for confounding factors, second-born twins showed an elevated likelihood of requiring supplemental oxygen, treatment with antibiotics and phototherapy, than first-born twin (Table 4).

## Discussion

The twin pregnancy rate has increased in the last three decades due to available technologies facilitating assisted reproduction, and because more women of advanced age ($\geq$ 35 years old) are becoming pregnant [7–9]. The proportion of twin pregnancies in this study was 1.15%, similar to results found in other studies in Brazil [12].

Twin pregnancies and births continue to present a challenge for health services. The risk of stillbirth is high, and the timing of delivery is important. Monochorionic twin pregnancies and dichorionic gestation, which often leads to early delivery, potentially increase the risk of

**Table 3. Twin pregnancy associated with neonatal and maternal outcomes by gestational age groups.**

| | Preterm (< 34 weeks) | | Late preterm (34–36 weeks) | | Early term (37–38 weeks) | | Full term (> = 39 weeks) | |
|---|---|---|---|---|---|---|---|---|
| | Crude OR (95%CI) | adj. OR* (95% CI) | Crude OR (95%CI) | adj. OR* (95% CI) | Crude OR (95%CI) | adj. OR* (95% CI) | Crude OR (95%CI) | adj. OR*(95% CI) |
| **Birth weight** | | | | | | | | |
| Low birth weight (< 2500 g) | NA NA | NA NA | 7.7 (4.3–13.7) | 7.4 (4.4–12.7) | 6.7 (3.9–11.7) | 6.6 (3.6–12.1) | 17.9 (5.8–55.4) | 14.8 (4.6–47.7) |
| SGA | 1.1 (0.5–2.4) | 0.9 (0.4–2.4) | 4.6 (2.1–11.0) | 4.3 (2.2–8.5) | 6.5 (3.8–11.3) | 6.3 (3.5–11.4) | 10.7 (4.3–26.5) | 12.2 (4.7–31.6) |
| IUGR | 1.0 (0.4–2.4) | 0.9 (0.3–2.7) | 3.5 (1.1–11.1) | 3.4 (1.3–8.6) | 7.2 (3.4–15.2) | 8.0 (3.7–17.3) | 17.7 (6.3–50.2) | 21.3 (7.1–64.2) |
| **Neonatal and maternal outcomes** | | | | | | | | |
| Resuscitation | 0.8 (0.4–1.6) | 0.6 (0.2–1.3) | 0.8 (0.4–2.8) | 0.8 (0.4–1.4) | 1.2 (0.5–2.9) | 1.3 (0.5–3.1) | 1.8 (0.4–7.5) | 2.0 (0.5–7.9) |
| Oxygen therapy | 3.4 (1.2–9.5) | 4.5 (1.6–12.1) | 1.3 (0.7–2.3) | 1.3 (0.7–2.4) | 3.0 (1.4–6.6) | 3.2 (1.5–7.0) | NA | NA |
| Antibiotic use | 1.1 (0.5–2.5) | 1.2 (0.5–2.8) | 1.8 (1.0–3.1) | 2.0 (1.1–3.6) | 2.4 (1.1–5.3) | 2.4 (1.1–5.6) | 0.6 (0.2–1.6) | 0.7 (0.2–1.9) |
| Neonatal ICU admission | 2.1 (1.0–4.5) | 2.5 (1.3–4.9) | 1.8 (1.2–2.8) | 1.6 (1.0–2.7) | 3.5 (1.7–7.3) | 3.8 (1.8–7.9) | NA | NA |
| Transient tachypnea | 1.6 (0.9–2.8) | 2.0 (1.1–3.7) | 1.3 (0.7–2.5) | 1.3 (0.6–2.7) | 4.1 (1.7–10.2) | 4.0 (1.5–10.5) | 1.1 (0.2–5.3) | 1.3 (0.3–5.7) |
| Hypoglycemia | 1.6 (0.7–3.5) | 1.8 (0.7–4.7) | 0.8 (0.3–2.3) | 0.8 (0.2–2.5) | 5.5 (1.7–16.6) | 6.2 (1.9–20.0) | NA | NA |
| Phototherapy (jaundice) | 2.0 (1.0–4.0) | 2.0 (0.9–4.4) | 3.4 (2.2–5.3) | 4.1 (2.7–6.2) | 3.3 (1.7–6.4) | 3.5 (1.7–7.0) | 2.20 (1.4–3.6) | 2.2 (1.3–3.8) |
| Severe Neonatal Outcome Index [a] | NA NA | NA NA | 2.4 (1.2–4.9) | 2.2 (1.2–3.8) | 1.8 (1.0–3.4) | 1.8 (0.9–3.3) | 2.02 (0.6–7.4) | 2.2 (0.6–8.5) |
| Severe Maternal Outcome Index [b] | NA NA | NA NA | 0.7 (0.2–2.5) | 0.3 (0.1–1.7) | 1.3 (0.2–8.8) | 1.2 (0.2–9.0) | NA | NA |

* Adjusted for region, type of payment for birth (public or private), mother's age, years of schooling and gestational age at birth in weeks' gestation.

a Neonatal nearmiss, stillbirth or neonatal death.

b Maternal nearmiss or maternal death.

NA: Logistic Regression not applicable due to very few, or too many, subjects with the outcome (see Table 2).

**Table 4. Twin pregnancy associated with neonatal and maternal outcomes by order of birth.**

| | All twins (n = 554) | Singletons (n = 23,746) | All twins (vs. Single) | Twin A (n = 277) | Twin B (n = 277) | Twin B (vs. Twin A) |
|---|---|---|---|---|---|---|
| | n (%) | n (%) | adj. OR* (95%CI) | n (%) | n (%) | OR** (95%CI) |
| **Birth weight** | | | | | | |
| Mean (SD) | 2240g (573g) | 3156g (573g) | NA | 2249g (558g) | 2193g (546g) | NA |
| Low birth weight (< 2500 g) | 353 (63.7) | 2116 (8.9) | 6.7 (4.6–9.9) | 176 (63.5) | 195 (70.4) | 1.0 (0.9–1.1) |
| SGA | 202 (36.5) | 1822 (7.7) | 7.3 (4.7–11.2) | 103 (37.2) | 99 (35.7) | 0.9 (0.8–1.1) |
| IUGR | 103 (18.6) | 746 (3.1) | 3.3 (1.9–5.8) | 51 (18.4) | 52 (18.8) | 1.0 (0.9–1.1) |
| **Neonatal and maternal outcomes** | | | | | | |
| Resuscitation | 83 (15.0) | 1769 (7.4) | 0.7 (0.4–1.4) | 35 (12.6) | 48 (17.3) | 1.5 (0.9–2.3) |
| Oxygen therapy | 161 (29.1) | 1538 (6.5) | 2.4 (1.0–5.5) | 72 (26.0) | 89 (32.1) | 1.3 (1.1–1.6) |
| Antibiotic use | 124 (22.4) | 1184 (5.0) | 1.6 (1.0–2.6) | 50 (18.1) | 74 (26.7) | 1.7 (1.3–2.1) |
| Neonatal ICU admission | 172 (31.0) | 1497 (6.3) | 3.8 (1.6–8.7) | 78 (28.2) | 94 (33.9) | 1.3 (0.9–2.1) |
| Transient tachypnea | 82 (14.8) | 773 (3.3) | 1.8 (1.1–2.9) | 36 (13.0) | 46 (16.6) | 1.3 (0.9–1.9) |
| Hypoglycemia | 33 (6.2) | 284 (1.2) | 1.4 (0.6–3.2) | 13 (4.7) | 20 (7.2) | 1.6 (0.7–3.6) |
| Phototherapy (jaundice) | 151 (27.3) | 1080 (4.5) | 3.4 (2.4–4.7) | 58 (20.9) | 93 (33.6) | 2.0 (1.5–2.6) |
| Stillbirth | 6 (1.1) | 263 (1.1) | 1.0 (0.99–1.01) | 3 (1.1) | 3 (1.1) | 1.0 (0.98–1.02) |
| Neonatal death | 27 (4.9) | 241 (1.0) | 4.6 (2.4–8.3) | 15 (5.4) | 12 (4.3) | 0.8 (0.48–1.16) |
| Severe Neonatal Outcome Index [a] | 247 (44.6) | 2852 (12.0) | 2.1 (1.4–3.3) | 117 (51.5) | 130 (57.3) | 1.2 (0.9–1.5) |
| Severe Maternal Outcome Index [b] | 8 (1.4) | 264 (1.1) | 0.3 (0.1–1.2) | NA | NA | NA |

* Adjusted for region, type of payment for birth (public or private), mother's age, years of schooling and gestational age at birth in weeks' gestation.

** There was no need for adjustment, since twins are from the same mother.

a Neonatal nearmisses, stillbitths or neonatal deaths.

b Maternal nearmisses (268 women) or maternal deaths (4 women).

NA: Logistic Regression not applicable.

neonatal complications [28–30]. Since the main objective of "Birth in Brazil" study was not to study twin births, it was not possible to determine whether the twin pregnancy was monochorionic or dichorionic. Neither was it possible to discern whether assisted reproductive technologies (ART) were applied during twin pregnancies. It was, however, possible to determine that the proportion of twins was higher in the private sector and in women >39 years, which increases the odds of twin pregnancy by ART. The same trend was observed in other studies in Brazil [12,31].

We found that the absolute difference between severe perinatal outcomes in twins compared to single born infants is higher during the late preterm period. The magnitude of this outcome reaches almost 50% of infants. The proportion of spontaneous and provider-initiated twin births was similar to single births in the late preterm; spontaneous births represented 60% of the total number of births. Thus, a possible explanation for the disparities of outcomes between twins and single born infants could be due to the higher prevalence of intrauterine growth restriction. Weight is a defining criterion for neonatal near miss and IUGR infants had greater occurrence of stillbirths. As for early term births, there was no significant increase in the chances of adverse perinatal outcomes of twins relative to single born infants. However, other neonatal outcomes— with the exception of resuscitation—were more prevalent in twins. These differences can be explained, in part, by the fact that 66% of births were provider-initiated, mainly prelabour caesarean, which increases the risk of respiratory morbidity of the newborn [32].

The optimal time of delivery for twin pregnancies is a highly debated topic [28–30,33]. A meta-analysis published in 2016 [30], which included 29,685 dichorionic and 5,486

monochronic pregnancies, showed that waiting for delivery beyond 37 weeks led to an additional 8.8/1,000 perinatal deaths in dichorionic twin pregnancies. For monochorionic twins, there was a non-significant trend towards an increase in stillbirths compared with neonatal deaths after 36 weeks. This analysis supported the notion that delivery should be carried out at 37 weeks for dichorionic pregnancies, and at 36 weeks for uncomplicated monochorionic twins. However, other authors support the close monitoring of twin pregnancies so as to avoid late preterm deliveries without increasing risk of stillbirth [34]. In our study, most twins were delivered between 34 and 38 weeks of gestational age (74.4%) and had significantly more neonatal complications associated with the gestational age (jaundice, neonatal ICU admission, hypoglycaemia, need for oxygen, and use of antibiotics, among other morbidities).

Even though we collected our data before new recommendations regarding the timing of delivery of twins up to 38 weeks were emphasised in literature [30,35], only 8% of twins were born after 38 weeks.

In our data, the non-significant difference in the odds of adverse perinatal outcomes between twins and early term single born infants suggests that early term delivery is more beneficial to infant and mother than late preterm delivery.

The mode of delivery observed in this study was predominantly caesarean as an obstetric intervention. This procedure continues to draw controversy, especially when applied to a second twin not in the cephalic position [17,18,36–38]. In a randomised control trial published in 2013, planned caesarean did not result in an increase or decrease of perinatal mortality or serious neonatal morbidity [18]. In a cohort study conducted in Australia [38], when comparing planned caesarean section with planned vaginal delivery in twin pregnancies with the first cephalic foetus, there was no difference in perinatal mortality, Apgar score < 4, and asphyxia-related morbidity. However, before 36 weeks and 6 days, planned caesarean section resulted in higher neonatal morbidity and mortality. After 37 weeks, planned caesarean section resulted in less asphyixia-related morbidity, but no difference in mortality and morbidity < 28 days, and Apgar < 4 [38]. Similar results for preterm deliveries were found in a French study, in which planned caesarean was associated with increased composite neonatal mortality and morbidity [37]. In this study, no difference was found for term deliveries. We found that for early term infants, two thirds of twins were delivered by obstetric intervention, mainly via planned caesarean. Although we did not find differences regarding severe perinatal outcomes, some of the differences found for the other outcomes may be due to the effect of prelabour caesarean section, such as a greater likelihood of transient tachypnoea, need for oxygen therapy, and neonatal ICU admission.

Unlike other studies [13,19,20], we found no differences in severe maternal outcomes in twin pregnancies compared to single pregnancies. One hypothesis for this result is that twin pregnancies may receive more prenatal care than single pregnancies and are referred for delivery in specialised referral services. Moreover, Madar et al. [20] recently found that one fifth of the association between twin pregnancy and severe maternal outcomes may be mediated by caesarean delivery, yet caesarean rate was also high for single births in our sample. Recent findings from a French study verified that caesarean for the second twin and for both twins had higher risk of severe maternal morbidity compared to vaginal delivery for both twins [39], which also emphasizes the importance of reducing caesarean rates for twins in Brazil.

## Conclusion

The twin pregnancy rate was similar to that found in other studies in Brazil. The proportion of caesarean sections was high, with 75% of newborns classified as late preterm and early term. Along with this came the inevitable greater occurrence of neonatal complications associated

with these gestational ages. However, all neonatal complications were more frequent in twins at all gestational ages, when compared to single births. Caesarean delivery may be the cause for poorer outcomes observed in early term twins.

## Supporting information

**S1 Fig. Distribution of gestational age at birth in singleton and twin infants.**
(TIF)

**S1 Table. Onset of labour and mode of birth in twin and singleton newborns by gestational age groups.**
(DOCX)

**S1 File.**
(ZIP)

## Author Contributions

**Conceptualization:** Ana Paula Esteves-Pereira, Antônio José Ledo Alves da Cunha, Marcos Nakamura-Pereira, Maria Elisabeth Moreira, Rosa Maria soares madeira Domingues, Maria do Carmo Leal, Silvana Granado nogueira da Gama.

**Data curation:** Ana Paula Esteves-Pereira.

**Formal analysis:** Ana Paula Esteves-Pereira, Antônio José Ledo Alves da Cunha, Marcos Nakamura-Pereira, Maria Elisabeth Moreira, Rosa Maria soares madeira Domingues, Elaine Fernandes Viellas, Maria do Carmo Leal, Silvana Granado nogueira da Gama.

**Funding acquisition:** Maria do Carmo Leal, Silvana Granado nogueira da Gama.

**Investigation:** Ana Paula Esteves-Pereira, Antônio José Ledo Alves da Cunha, Marcos Naka-mura-Pereira, Maria Elisabeth Moreira, Rosa Maria soares madeira Domingues, Elaine Fer-nandes Viellas, Maria do Carmo Leal, Silvana Granado nogueira da Gama.

**Methodology:** Ana Paula Esteves-Pereira, Antônio José Ledo Alves da Cunha, Marcos Naka-mura-Pereira, Rosa Maria soares madeira Domingues, Maria do Carmo Leal, Silvana Gran-ado nogueira da Gama.

**Writing – original draft:** Ana Paula Esteves-Pereira, Antônio José Ledo Alves da Cunha, Mar-cos Nakamura-Pereira, Maria Elisabeth Moreira, Rosa Maria soares madeira Domingues, Elaine Fernandes Viellas, Maria do Carmo Leal, Silvana Granado nogueira da Gama.

**Writing – review & editing:** Ana Paula Esteves-Pereira, Antônio José Ledo Alves da Cunha, Marcos Nakamura-Pereira, Maria Elisabeth Moreira, Rosa Maria soares madeira Domin-gues, Elaine Fernandes Viellas, Maria do Carmo Leal, Silvana Granado nogueira da Gama.

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
