## [Editor Report · Decision Letter 0]

13 Oct 2020

PONE-D-20-23490

Twin pregnancy and perinatal outcomes: data from ‘Birth in Brazil Study’

PLOS ONE

Dear Dr. Esteves-Pereira,

Thank you for submitting your manuscript to PLOS ONE. After careful consideration, we feel that it has merit but does not fully meet PLOS ONE’s publication criteria as it currently stands. Therefore, we invite you to submit a revised version of the manuscript that addresses the points raised during the review process.

We look forward to receiving your revised manuscript.

Kind regards,

Andrew Sharp, PhD

Academic Editor

PLOS ONE

Journal Requirements:

3. Please amend your manuscript to include your abstract after the title page.

Additional Editor Comments:

A large cohort study of twins and singletons in brazil. The authors have attempted to compare the outcomes of twins to singletons at a variety of gestational age brackets.

Likely, the increased age and hypertensive disorders are linked to assisted reproduction but the authors did not assess this. This would fit with others studies in twins. Likewise chorionicity was not assessed which is a common failing of large cohorts of twins

The lack of detail about the incidence of preterm birth <34 weeks for twins is a shame as this is one of the highest risk groups. In addition, it is unclear which proportion of these babies in each group were delivered due to other clinical concerns or who laboured spontaneously. The approach to interpreting the data would be significantly different if the cause were not potentially preventable, i.e. delivering due to TTTS or severe IUGR in which case you would accept the risks of prematurity whatever they are

On page 5 please make the definition of IUGR and SGA clearer as <3rd and <10th respectively. You do this in the tables so I suspect you are using the Delphi criteria but the text does not make this clear

Page 5 skin colour is an unusual term to use for ethnicity for this reviewer. I would suggest changing this to ethnicity to improve the translation to other readers.

Table 1 can you split the mode of delivery into elective (no labour) CS and emergency CS. It is unclear at present

Please add data for stillbirths and neonatal deaths to all relevant tables
---

## [Author Response · Author response to Decision Letter 0]

1 Dec 2020

Manuscript “Twin pregnancy and perinatal outcomes: data from ‘Birth in Brazil Study” 

[PONE-D-20-23490].

Each point raised has been answered as follows:

Response: The manuscript was amended to meet PLOS ONE's style requirements, including those for file naming.

2. We note that you have indicated that data from this study are available upon request. PLOS only allows data to be available upon request if there are legal or ethical restrictions on sharing data publicly. 

Response: The authors have upload the minimal anonymized data set necessary to replicate the study findings as a Supporting Information file “NB1_GEMEOS_PLOSONE.sav”

3. Please amend your manuscript to include your abstract after the title page.

Response: The abstract (now page 2) was included in the manuscript after the title page.

A large cohort study of twins and singletons in brazil. The authors have attempted to compare the outcomes of twins to singletons at a variety of gestational age brackets.

4.1. Likely, the increased age and hypertensive disorders are linked to assisted reproduction but the authors did not assess this. This would fit with others studies in twins. Likewise chorionicity was not assessed which is a common failing of large cohorts of twins.

Response: Since the main objective of “Birth in Brazil” study was not to study twin births, it was not possible to determine whether the twin pregnancy was monochorionic or dichorionic. Neither was it possible to discern which woman used assisted reproductive technologies (ART). This is a limitation of the study, addressed in the discussion section. 

However, we described that the proportion of twins was higher in the private sector and in women >39 years, which may be due, at least in some extent, to using ART more frequently than younger women from the public sector do. 

We will be able to fully address this hypothesis in Birth in Brazil 2, with data collection beginning in 2021.

4.2.The lack of detail about the incidence of preterm birth <34 weeks for twins is a shame as this is one of the highest risk groups. In addition, it is unclear which proportion of these babies in each group were delivered due to other clinical concerns or who laboured spontaneously. The approach to interpreting the data would be significantly different if the cause were not potentially preventable, i.e. delivering due to TTTS or severe IUGR in which case you would accept the risks of prematurity whatever they are.

Response: The authors have included two supporting files (Supplemental_Table_1 and Suplemental_Figure_1), which are described in the results section (page 9). These files show the detail about the incidence of preterm birth <34 weeks for twins and singletons, as well as the proportion of newborns with provider-initiated onset of labour (induction or elective caesarean) or who laboured spontaneously.

4.3. On page 5 please make the definition of IUGR and SGA clearer as <3rd and <10th respectively. You do this in the tables so I suspect you are using the Delphi criteria but the text does not make this clear.

Response: In now page 6, we changed the original text to: “Appropriate birth weight for gestational age was assessed by Intergrowth-21st intrauterine growth curves, considering birth weight <10th percentile as small for gestational age (SGA) and birth weight <3th percentile as intrauterine growth restriction (IUGR).”

4.4. Page 5 skin colour is an unusual term to use for ethnicity for this reviewer. I would suggest changing this to ethnicity to improve the translation to other readers.

Response: The term “skin colour” was changed for “ethnicity” in the text and tables.

4.5. Table 1 can you split the mode of delivery into elective (no labour) CS and emergency CS. It is unclear at present.

Response: Table 1 (page 10) and its results (page 9) were updated as suggested.

4.6. Please add data for stillbirths and neonatal deaths to all relevant tables

Response: The data for stillbirths and neonatal deaths were included in tables 2 and 4.

---

## [Editor Report · Decision Letter 1]

23 Dec 2020

Twin pregnancy and perinatal outcomes: data from ‘Birth in Brazil Study’

PONE-D-20-23490R1

Dear Dr. Esteves-Pereira,

We’re pleased to inform you that your manuscript has been judged scientifically suitable for publication and will be formally accepted for publication once it meets all outstanding technical requirements.

Kind regards,

Andrew Sharp, PhD

Academic Editor

PLOS ONE
---

## [Editor Report · Acceptance letter]

30 Dec 2020

PONE-D-20-23490R1 

Twin pregnancy and perinatal outcomes: data from ‘Birth in Brazil Study’ 

Dear Dr. Esteves-Pereira:

I'm pleased to inform you that your manuscript has been deemed suitable for publication in PLOS ONE. Congratulations! Your manuscript is now with our production department. 

Kind regards, 

on behalf of

Dr. Andrew Sharp 

Academic Editor

PLOS ONE